# Monthly Disposable Income Is a Crucial Factor Affecting the Quality of Life in Patients with Knee Osteoarthritis

**DOI:** 10.3390/healthcare9121703

**Published:** 2021-12-08

**Authors:** Tian-Shyug Lee, Hsiang-Chuan Liu, Wei-Guang Tsaur, Shih-Pin Lee

**Affiliations:** 1Graduate Institute of Business Administration, Fu Jen Catholic University, No. 510, Zhongzheng Rd., Xinzhuang Dist., New Taipei City 24205, Taiwan; 036665@mail.fju.edu.tw (T.-S.L.); fpan0366@yahoo.com.tw (H.-C.L.); 2Department of Economics, Fu Jen Catholic University, New Taipei City 24205, Taiwan; 003397@mail.fju.edu.tw

**Keywords:** demographic characteristics, quality of life, physical health, mental health, monthly disposable income

## Abstract

Knee osteoarthritis (OA) affects the quality of life (QOL) of elderly people; this study examines the demographic characteristics and QOL of patients with knee OA and identifies demographic characteristics that affect the QOL of these patients. In this cross-sectional study, 30 healthy controls and 60 patients with mild-to-moderate bilateral knee OA aged between 55 and 75 years were enrolled. All participants completed a questionnaire containing questions on 10 demographic characteristics and the Medical Outcome Study 36-Item Short-Form Health Survey (SF-36), and their QOL scores in the eight dimensions of the SF-36 were evaluated. In the OA group, significant correlations were observed between monthly disposable income and physical and mental health components. Monthly disposable income was found to considerably affect the QOL of patients with bilateral knee OA (i.e., it is a crucial factor affecting these patients). The findings of this study may provide a reference for formulating preventive strategies for healthy individuals and for future confirmatory research.

## 1. Introduction

The increase in the aging population globally has led to an increase in the prevalence of chronic diseases, including osteoarthritis (OA), dementia, stroke, and coronary heart disease, among older people [1]. OA is a degenerative and progressive joint disease that mainly involves weight-bearing joints, such as the hip, knee, and ankle joints, and it is considered one of the leading causes of lower limb disabilities in older people [2].

Knee OA, one of the most common forms of OA worldwide, can lead to the severe loss of joint function [3], restricted activities of daily living [4], and disability—all of which may pose a considerable socioeconomic burden on patients, their families, and society [2,3]. By 2025, the worldwide prevalence of age-related knee OA is estimated to increase by 40% [5]. It is estimated that about 10% of the population over 60 years of age complain of knee pain, disability, functional impairment, and a corresponding diminished quality of life (QOL) [6]. Approximately 37% of the population over 60 years old in the US have a diagnosis of knee OA [7]. Together, discomfort and limited range of motion worsen the QOL of older people with knee OA.

The aforementioned decline in the activities of daily living caused by knee OA progression considerably deteriorates patients’ work, social life, sleep quality, and, consequently, overall QOL [8,9]. Therefore, assessing QOL is essential when evaluating patients with knee OA. The World Health Organization defines QOL as “an individual’s perception of their position in life in the context of the culture and value systems in which they live and in relation to their goals, expectations, standards, and concerns” [10].

The Medical Outcomes Study 36-Item Short-Form Health Survey (SF-36), a coherent and universal assessment tool, measures QOL by considering factors such as psychological and social well-being. Thus, the SF-36 can be used to evaluate the QOL of knee OA patients with comorbidities [11]. Many studies have investigated patients’ symptoms only from the perspective of pathology, neglecting the importance of patients’ subjective perceptions and feelings. Currently, treatment effectiveness in patients with knee OA, whether for research or medical purposes, lacks a subjective self-assessment of patients’ QOL [12,13]. Traditional medical rehabilitation prescriptions only focus on enhancing knee movement and alleviating peripheral edema and pain [14,15,16]. Therefore, treatment objectives often do not consider patients’ comprehensive health, subjective perception of the disease, and health-related QOL.

The SF-36 measures patients’ opinions regarding their lives and physical activities [13]. Therefore, in addition to improving the symptoms of knee OA as the main treatment goal, the current medical practice focuses on maintaining QOL after the onset of illness. Many clinical studies have considered health-related QOL as an essential variable for evaluating therapeutic effects. In addition, many studies have reported the trend of SF-36 scores in patients with knee OA after treatment [17,18,19,20]. However, whether similarities in demographic characteristics among the subgroups of the study population affect the trend of SF-36 scores in these patients has rarely been investigated; the aforementioned demographic characteristics include age, sex, socioeconomic status, insurance coverage, educational level, monthly disposable income, and socioeconomic status. This consideration is crucial because if no between-group differences are noted in the aforementioned characteristics, the true between-group differences in SF-36 scores may be masked, with deviations still existing at the individual level [21].

Most studies examining the QOL of patients with knee OA have investigated only symptoms that affect the knee joint [11,17,18,19,20,22]; few studies have explored demographics and QOL. Therefore, the present study examined the demographic characteristics and QOL of patients with knee OA and identified demographic characteristics that affect the QOL of these patients.

## 2. Materials and Methods

### 2.1. Participants

In this cross-sectional study, 60 older patients aged between 55 and 75 years with mild-to-moderate bilateral knee OA (grade 2 or 3 on the Kellgren–Lawrence scale (KL scale)) who received treatment in a rehabilitation clinic were recruited through simple random sampling among those who met the study’s inclusion criteria. In addition, 30 age-matched healthy controls were included in this study. Some controls were selected after responding to the advertisement for the research. Additional controls were recruited from the families of patients who visited the clinic during the research period.

The same physician confirmed the diagnosis of knee OA through X-ray (weight-bearing anteroposterior, lateral, and skyline views) by using the KL scale and assisted in confirming the general health of enrolled participants. According to the KL scale, the knees that showed no features of OA were assigned grade 0. The knees that exhibited joint space narrowing and possible osteophytes were suspected to have OA and were assigned grade 1. The knees that showed small osteophytes and possible joint space narrowing were classified as having mild OA and were assigned grade 2. The knees that exhibited multiple, moderately sized osteophytes, definite joint space narrowing, and possible bony end deformity were classified as having moderate OA and were assigned grade 3. Finally, the knees showing multiple large osteophytes, severe joint space narrowing, marked sclerosis, and definite bony end deformity were classified as having severe OA and were assigned grade 4 [23].

After assessments by a neurologist and a physical therapist, patients who had severe knee OA that caused difficulty in standing or other major injuries and illnesses affecting the study outcome, including an American Society of Anesthesiologists grade of ≥2 for cardiopulmonary function, neurological abnormalities, cardiopulmonary failure, and a history of stroke, were excluded. In addition, patients participating in other studies were excluded. Controls reported no current or past lower limb pain, the physical examination of both knees were normal, and the self-reported history of vertigo, stroke, or other conditions that might impair balance were excluded.

All patients provided informed consent for study participation, and their demographic data were collected. This study was approved by the Ethics Committee of Fu Jen Catholic University (FJU-IRB NO: C107179).

### 2.2. Procedure

All patients completed a questionnaire containing the written consent form, questions related to basic demographic information, and the SF-36. The demographic characteristics of participants included age, sex, height, weight, history of chronic disease, low-income household, marital status, monthly disposable income, educational level, number of insurance policies, and pain scale score (visual analog scale (VAS)).

The number of chronic diseases ranged from 0 to 3; the eligibility of a low-income household was determined in accordance with Taiwan’s New Taipei City Government in 2021: the total household income distribution for the whole family is 1.5 times lower than the minimum living allowance per person per month, which is NT$23,400, which is about US$836; marital status was divided into unmarried, widowed, and married; monthly disposable income represents the approximate amount of money that needs to be spent in daily life every month; the education level was divided into ≤9 years, 9–12 years, and >12 years of education; the number of insurance policies included at least the National Health Insurance in Taiwan (≤1) or reinsurance from other insurance companies (>1); and the knee pain scale evaluated the intensity of knee pain experienced during walking. The intensity of knee pain was scored on a 10 cm horizontal VAS marked in 1 cm increments, with a score of 0 cm indicating “no pain” and a score of 10 cm indicating “pain as bad as it could be” or “worst imaginable pain”. The VAS score of knee pain during walking was recorded at the preferred walking speed on an even level in an outdoor area [24].

The SF-36 adopts the Chinese version of SF-36 established by Liu et al. [25], and the score calculation method adopts the scoring method of McHorney et al. [26]. The SF-36 has two major components, each of which has four dimensions: the physical health component (four dimensions: physical function (PF), body role/role limitations due to physical health problems (BR), body pain (BP), and general health problems (GH)) and the mental health component (four dimensions: vitality (VT), social function (SF), emotional status/role limitations due to emotional health problems (ES), and general mental health problems (MH)). Eight dimensions were evaluated in total, and for each dimension, we obtained a score after applying a measurement scale ranging from 0 (poorest health status) to 100 (most favorable health status) [11]. The SF-36 was completed by the subjects without assistance.

We used 10 demographic characteristics, namely age, sex, height, weight, history of chronic disease, low-income household, marital status, monthly disposable income, educational level, and number of insurance policies, to compare the eight dimensions of the SF-36 (PF, BR, BP, GH, VT, SF, ES, and MH) and to identify demographic characteristics affecting QOL in the control and OA groups. Subsequently, we used the aforementioned demographic characteristics to compare the physical and mental health components of the SF-36 for determining demographic characteristics affecting the physical and mental health components in the control and OA groups.

### 2.3. Statistical Analysis

All statistical analyses were performed using R software (version 3.6.1; R Foundation for Statistical Computing, Vienna, Austria). Descriptive statistics were generated for the demographic characteristics of the control and OA groups. Furthermore, we calculated the means and standard deviations of continuous variables including age, height, weight, monthly disposable income, and VAS scores; Mann–Whitney U tests were used to compare whether there were statistical differences between the control and OA groups. The counts and percentages of categorical variables included sex, history of chronic disease, low-income household, marital status, educational level, and number of insurance policies; a Chi-square test was used to test whether there were statistical differences. A multiple regression analysis was performed to evaluate the associations of the demographic characteristics with the SF-36, physical health components, and mental health components in the control and OA groups. For all analyses, a *p* value of <0.05 indicated statistical significance.

## 3. Results

Table 1 presents the comparison of the demographic characteristics between the control (n = 30) and OA (n = 60) groups. No significant differences in age (*p* = 0.920), sex (*p* = 0.940), height (*p* = 0.249), weight (*p* = 0.713), history of chronic disease (*p* = 0.380), or low-income household (*p* = 0.097) were observed between the OA and control groups. However, significant differences in marital status (*p* = 0.011), monthly disposable income (*p* < 0.001), educational level (*p* = 0.040), and number of insurance policies (*p* = 0.014) were found between the control and OA groups.

Table 2 lists the results of the QOL assessment based on the physical health component (PF, BR, BP, and GH) and mental health component (VT, SF, ES, and MH) of the SF-36 between the control and OA groups. All the variables significantly differed between the control and OA groups (*p* < 0.001), indicating the superior physical and mental health of the control group.

Table 3 presents the results of the multiple regression analysis of the association between demographic characteristics and the average SF-36 score for the eight dimensions in the control and OA groups. In the control group, no significant difference was noted between SF-36 scores and demographic characteristics, namely, age (*p* = 0.241), sex (*p* = 0.401 (reference group = male)), height (*p* = 0.325), weight (*p* = 0.819), history of chronic disease (*p* = 0.149), low-income household (*p* = 0.269 (reference group = not low)), marital status as widowed (*p* = 0.874), marital status as married (*p* = 0.960 (reference group = unmarried)), monthly disposable income (*p* = 0.113), education for 9–12 years (*p* = 0.724), education for >12 years (*p* = 0.239 (reference group = education for ≤9 years)), and number of insurance policies (*p* = 0.315). In the OA group, monthly disposable income (*p* = 0.013) and number of insurance policies (*p* = 0.046) were significantly associated with SF-36 scores. However, in the OA group, age (*p* = 0.272), sex (*p* = 0.684 (reference group = male)), height (*p* = 0.864), weight (*p* = 0.858), history of chronic disease (*p* = 0.312), low-income household (*p* = 0.291 (reference group = not low), marital status as widowed (*p* = 0.219), marital status as married (*p* = 0.387 (reference group = unmarried)), education for 9–12 years (*p* = 0.510), and education for >12 years (*p* = 0.930 (reference group = education for ≤9 years)) were not associated with SF-36 scores.

We compared the association of demographic characteristics with the average scores of the physical health and mental health components of the SF-36. The association between demographic characteristics and physical health components is presented in Table 4. In the control group, no significant differences were observed between physical health components and age (*p* = 0.100), sex (*p* = 0.865 (reference group = male)), height (*p* = 0.322), weight (*p* = 0.779), history of chronic disease (*p* = 0.371), low-income household (*p* = 0.233 (reference group = not low)), marital status as widowed (*p* = 0.749), marital status as married (*p* = 0.524 (reference group = unmarried)), monthly disposable income (*p* = 0.125), education for 9–12 years (*p* = 0.865), education for >12 years (*p* = 0.480 (reference group = education for ≤9 years)), and number of insurance policies (*p* = 0.896). In the OA group, only monthly disposable income (*p* = 0.020) was significantly associated with physical health components. Age (*p* = 0.115), sex (*p* = 0.413 (reference group = male)), height (*p* = 0.791), weight (*p* = 0.814), history of chronic disease (*p* = 0.265), low-income households (*p* = 0.232 (reference group = not low)), marital status as widowed (*p* = 0.445), marital status as married (*p* = 0.840 (reference group = unmarried)), education for 9–12 years (*p* = 0.397), education for >12 years (*p* = 0.958 (reference group = education for ≤9 years)), and number of insurance policies (*p* = 0.210) were not associated with the physical health components in the OA group.

Table 5 presents the association between demographic characteristics and mental health components. In the control group, no significant differences were observed between mental health components and age (*p* = 0.970), sex (*p* = 0.178 (reference group = male)), height (*p* = 0.560), weight (*p* = 0.947), history of chronic disease (*p* = 0.114), low-income household (*p* = 0.584 (reference group = not low)), marital status as widowed (*p* = 0.477), marital status as married (*p* = 0.359 (reference group = unmarried)), monthly disposable income (*p* = 0.303), education for 9–12 years (*p* = 0.378), education for >12 years (*p* = 0.188 (reference group = education for ≤9 years)), and number of insurance policies (*p* = 0.092). In the OA group, monthly disposable income (*p* = 0.014) and number of insurance policies (*p* = 0.013) were significantly associated with mental health components. However, age (*p* = 0.527), sex (*p* = 0.967 (reference group = male)), height (*p* = 0.933), weight (*p* = 0.609), history of chronic disease (*p* = 0.386), low-income household (*p* = 0.379 (reference group = not low)), marital status as widowed (*p* = 0.123), marital status as married (*p* = 0.174 (reference group = unmarried)), education for 9–12 years (*p* = 0.641), and education for >12 years (*p* = 0.842 (reference group = education for ≤9 years)) were not associated with the mental health components in the OA group.

## 4. Discussion

Studies on knee OA and QOL [11,27] have indicated that risk factors for knee OA include female sex, older age, low socioeconomic status, and low educational level. The results of our study revealed that a higher proportion of older patients with knee OA had a lower monthly disposable income, lower educational level, and fewer insurance policies (Table 1). In addition, most of these patients were widowed. This finding is in accordance with that reported by Jiao et al. [28], who indicated that long-term married participants had a higher income, more satisfactory healthy habits, and a higher QOL than did widowed participants.

Knee OA is a common degenerative disease in older adults that causes pain, stiffness, and dysfunction. Many studies have reported that knee OA reduces QOL [11,22,29]. To examine whether demographic characteristics affect the SF-36 scores of patients with knee OA, in this study, we examined the association of 10 demographic characteristics, namely, age, sex, height, weight, history of chronic disease, low-income household, marital status, monthly disposable income, educational level, and number of insurance policies with the eight dimensions (PF, BR, BP, GH, VT, SF, ES, and MH) of the SF-36 and identified the demographic characteristics that affect QOL.

We performed a statistical analysis to compare the control and OA groups in terms of the demographic characteristics and QOL. The results revealed that monthly disposable income and number of insurance policies were significantly associated with QOL in the OA group. Furthermore, to explore the relationship of demographic characteristics with physical and mental health components, we analyzed the associations of the demographic characteristics with the physical and mental health components in the OA group and compared them with those in the control group. The results revealed a significant association of monthly disposable income with physical and mental health components. The findings of the comparison between the demographic characteristics and mental health components in the OA group revealed that monthly disposable income and number of insurance policies were correlated with mental health components. Monthly disposable income exerted a stronger effect than the number of insurance policies on the mental health components (Table 5, *β*: 0.343 > 0.331) in the OA group.

In the OA group, monthly disposable income was significantly correlated with QOL, indicating that a higher QOL score corresponded to a higher monthly disposable income. Overall, in this study, monthly disposable income was a key factor affecting QOL. Solmi et al. described the complex interactions of a multidimensional set of variables in North American adults with a risk of knee OA and indicated that economic ability and physical and mental health-related QOL were closely related; this is in agreement with the finding of the present study that monthly disposable income affected physical and mental health [30].

Higher monthly disposable income was associated with a higher QOL score. Previous studies have shown that in patients and healthy individuals, a better physical health component of QOL was associated with a higher monthly disposable income [31]. Costa and Nogueira [31] reported that QOL was affected by factors associated with socioeconomic status and individual characteristics, such as income, educational level, and occupation, which are the determinants of an individual’s health and crucial in disease prevention and health intervention planning.

Indeed, according to a study of chronic illnesses, after the family’s basic living and health care expenditures, if the family’s “discretionary” income is too low, it can hardly meet the medical expenditures, which will affect health [32], and it will also cause a heavy burden on the patient’s physical and mental health. In a study reviewing the existing data on the epidemiology of OA and the burden of the disease, it was shown that the burden of OA is physical, psychological, and socioeconomic [33]. By focusing on the burden of this prevalent, disabling, and costly disease, Hunter et al. emphasized the opportunity for a shift in healthcare policy towards prevention and chronic disease management [34]. Similarly, in China, OA not only imposes a heavy burden on the population, but also affects gross domestic production (GDP). Therefore, scholars have promoted the urgent need for health policy support and cost-effective preventive strategies in China [35].

The results of the present study revealed that the monthly disposable income of patients with knee OA was closely related to their physical and mental health. For developing strategies to improve the rehabilitation of knee OA in older patients, follow-up studies should be conducted to further examine the effects of personal monthly disposable income, QOL, despair, and obesity on positive coping strategies adopted in the process of disease progression. Moreover, the aforementioned factors can be used to implement preventive strategies for knee OA in older patients. Even if knee OA and its determinants of demographic-related QOL are common in the general population, other studies are required to apply our results to the general population or different subpopulations of interest.

### Study Limitations

This study included patients from only one clinic, limited to one area, and the recruitment age was limited to 55–75 years old; thus, it would be difficult to promote the results to other area or groups of other ages. In addition, this cross-sectional observational study had a small sample size; thus, the data may not be sufficient for analyzing changes over time or for making causal inferences. However, the aim of this study was to identify the determinants of demographic-related QOL in patients with knee OA. The findings of this study may provide a basis for formulating preventive strategies and can provide a reference basis for future confirmatory research.

## 5. Conclusions

The results of this study demonstrated that monthly disposable income is a crucial factor affecting QOL and may provide a reference basis for formulating preventive strategies and for future confirmatory research. For earlier intervention in patients with knee OA and future related research, this study provides the following recommendations:Aside from demographic characteristics, pathological symptoms, irrespective of whether patients underwent surgery and their physical activity and career, can be relevant factors affecting the QOL of patients with knee OA. These factors should be examined in future studies.Follow-up studies should be conducted to develop strategies for improving the rehabilitation of knee OA in older patients, and positive coping strategies should be adopted in the process of disease progression.Monthly disposable income is a crucial factor affecting QOL. Thus, patients and healthy individuals should plan their careers early.Previous studies discussing the mental health of patients with knee OA are lacking. Thus, future studies should focus on improving the mental health of these patients.

## Figures and Tables

**Table 1 healthcare-09-01703-t001:** Demographic characteristics of patients in the control and osteoarthritis groups.

Demographic Characteristics	Item	Control Group (n = 30)	OA Group (n = 60)	*p*-Value
Age (years) (SD)	-	66.40 (5.48)	66.28 (4.98)	0.784
Sex (number) (%)	Male	17 (56.7)	32 (53.3)	0.940
	Female	13 (43.3)	28(46.7)	
Height (cm) (SD)	-	162.03 (6.11)	160.40 (6.38)	0.224
Weight (kg) (SD)	-	70.03 (8.62)	69.33 (8.42)	0.485
History of chronic disease (number) (%)	0 type	7 (23.3)	13 (21.7)	0.380
	1 type	18 (60.0)	28 (46.7)	
	2 types	5 (16.7)	16 (26.7)	
	3 types	0 (0.0)	3 (5.0)	
Low-income households (number) (%)	0 (Not low)	24 (80.0)	36 (60.0)	0.097
	1 (Low)	6 (20.0)	24 (40.0)	
Marital status (number) (%)	Unmarried	3 (10.0)	6 (10.0)	0.011
	Widowed	8 (26.7)	35 (58.3)	
	Married	19 (63.3)	19 (31.7)	
Monthly disposable income (million NTD) (SD)	-	2.87 (1.21)	1.83 (0.74)	<0.001
Education (number) (%)	Less than 9 years (≤9)	12 (40.0)	37 (61.7)	0.040
	9–12 years	11 (36.7)	19 (31.7)	
	More than 12 years (>12)	7 (23.3)	4 (6.7)	
Number of insurances policies (number) (%)	≤1	8 (26.7)	34 (56.7)	0.014
	>1	22 (73.3)	26 (43.3)	
VAS (mean) (SD)	-	NaN (NA)	7.35 (1.01)	NA

Statistically significant mean difference (*p* < 0.05). Values are expressed as the mean (SD: standard deviation).

**Table 2 healthcare-09-01703-t002:** Mean and standard deviation of the SF-36 in the control and osteoarthritis groups.

Components	Dimensions	Control Group (n = 30)	OA Group (n = 60)	*p*-Value
Physical health component	PF	91.25 (11.54)	36.50 (24.28)	<0.001
	BR	84.58 (23.31)	28.33 (24.99)	<0.001
	BP	82.08 (26.40)	17.29 (19.81)	<0.001
	GH	75.50 (15.50)	36.83 (21.35)	<0.001
Mental Health component	VT	68.54 (17.17)	41.25 (19.23)	<0.001
	SF	87.92 (15.22)	36.46 (32.81)	<0.001
	ES	72.22 (27.88)	28.75 (26.81)	<0.001
	MH	69.26 (14.35)	44.44 (24.33)	<0.001

Statistically significant mean difference (*p* < 0.05). Values are expressed as the mean (SD: standard deviation). PF: Physical function; BR: role limitations due to physical health problems; BP: body pain; GH: general health problems; VT: vitality; SF: social function; ES: role limitations due to emotional health problems; MH: general mental health problems.

**Table 3 healthcare-09-01703-t003:** Multiple regression analysis between demographic characteristics and SF-36 in control and osteoarthritis groups.

Demographic Characteristics		Control Group (n = 30) Adj-R^2^ = 0.196				OA Group (n = 60) Adj-R^2^ = 0.104			
		Unstandardized Coefficients		*β*	*p*-Value	Unstandardized Coefficients		*β*	*p*-Value
		B	95% CI			B	95% CI		
Age		0.549	−0.404, 1.502	0.277	0.241	−0.926	−2.603, 0.751	−0.244	0.272
Sex = Female (RG = Male)		−5.426	−18.713, 7.861	−0.251	0.401	−2.670	−15.785, 10.445	−0.071	0.684
Height		0.509	−0.550, 1.569	0.286	0.325	−0.076	−0.966, 0.813	−0.026	0.864
Weight		−0.074	−0.749, 0.601	−0.059	0.819	0.066	−0.670, 0.802	0.029	0.858
History of chronic disease		−6.028	−14.446, 2.390	−0.355	0.149	4.412	−4.278, 13.102	0.191	0.312
Low-income households (RG = Not low)		5.483	−4.632, 15.599	0.205	0.269	−5.753	−16.593, 5.088	−0.150	0.291
Marital status (RG = Unmarried)	Widowed	−1.433	−20.263, 17.397	−0.059	0.874	−12.810	−33.501, 7.882	−0.337	0.219
	Married	−0.353	−14.911, 14.206	−0.016	0.960	−8.820	−29.126, 11.486	−0.219	0.387
Monthly disposable income		3.728	−0.979, 8.436	0.415	0.113	8.860	1.926, 15.794	0.347	0.013
Education (RG = Education ≤ 9 years)	9–12 years	−1.802	−12.387, 8.782	−0.081	0.724	−4.880	−19.665, 9.904	−0.121	0.510
	>12 years	−7.390	−20.155, 5.374	−0.292	0.239	−0.907	−21.524, 19.709	−0.012	0.930
Number of insurances policies		−6.968	−21.170, 7.234	−0.288	0.315	−10.060	−19.928, −0.193	−0.266	0.046

Statistically significant mean difference (*p* < 0.05). B: Regression coefficient; CI: confidence intervals; *β*: standardized coefficients; Adj-R^2^: adjusted R squared.

**Table 4 healthcare-09-01703-t004:** Multiple regression analysis between demographic characteristics and physical health component in control and osteoarthritis groups.

Demographic Characteristics		Control Group (n = 30) Adj-R^2^ = 0.110				OA Group (n = 60) Adj-R^2^ = 0.083			
		Unstandardized Coefficients		*β*	*p*-Value	Unstandardized Coefficients		*β*	*p*-Value
		B	95% CI			B	95% CI		
Age		1.080	−0.229, 2.389	0.417	0.100	−1.252	−2.819, 0.316	−0.356	0.115
Sex = Female (RG = Male)		−1.493	−19.756, 16.770	−0.053	0.865	−5.033	−17.295, 7.229	−0.145	0.413
Height		0.703	−0.753, 2.160	0.302	0.322	−0.110	−0.942, 0.721	−0.040	0.791
Weight		−0.125	−1.053, 0.802	−0.076	0.779	−0.081	−0.769, 0.607	−0.039	0.814
History of chronic disease		−5.037	−16.607, 6.533	−0.227	0.371	4.557	−3.568, 12.682	0.214	0.265
Low-income households (RG = Not low)		8.143	−5.760, 22.046	0.233	0.233	−6.106	−16.242, 4.029	−0.173	0.232
Marital status (RG = Unmarried)	Widowed	3.987	−21.895, 29.868	0.126	0.749	−7.406	−26.752, 11.940	−0.211	0.445
	Married	6.172	−13.839, 26.182	0.213	0.524	−1.918	−20.903, 17.067	−0.051	0.840
Monthly disposable income		4.953	−1.517, 11.422	0.422	0.125	7.733	1.250, 14.216	0.327	0.020
Education (RG = Education ≤ 9 years)	9–12 years	1.192	−13.356, 15.739	0.041	0.865	−5.872	−19.695, 7.951	−0.158	0.397
	>12 years	−6.009	−23.554, 11.535	−0.182	0.480	0.503	−18.773, 19.778	0.007	0.958
Number of insurances policies		−1.228	−20.748, 18.292	−0.039	0.896	−5.831	−15.057, 3.394	−0.167	0.210

Statistically significant mean difference (*p* < 0.05). B: Regression coefficient; CI: confidence intervals; *β*: standardized coefficients; Adj-R^2^: adjusted R squared.

**Table 5 healthcare-09-01703-t005:** Multiple regression analysis between demographic characteristics and mental health component in control and osteoarthritis groups.

Demographic Characteristics		Control Group (n = 30) Adj-R^2^ = 0.164				OA Group (n = 60) Adj-R^2^ = 0.123			
		Unstandardized Coefficients		*β*	*p*-Value	Unstandardized Coefficients		*β*	*p*-Value
		B	95% CI			B	95% CI		
Age		0.018	−0.989, 1.025	0.009	0.970	−0.600	−2.493, 1.293	−0.138	0.527
Sex = Female (RG = Male)		−9.358	−23.398, 4.682	−0.419	0.178	−0.307	−15.110, 14.497	−0.007	0.967
Height		0.315	−0.804, 1.435	0.171	0.560	−0.042	−1.046, 0.962	−0.012	0.933
Weight		−0.023	−0.736, 0.690	−0.017	0.947	0.213	−0.618, 1.043	0.083	0.609
History of chronic disease		−7.019	−15.914, 1.876	−0.399	0.114	4.268	−5.541, 14.077	0.162	0.386
Low-income households (RG = Not low)		2.824	−7.864, 13.513	0.102	0.584	−5.399	−17.635, 6.837	−0.124	0.379
Marital status (RG = Unmarried)	Widowed	−6.853	−26.750, 13.044	−0.274	0.477	−18.213	−41.568, 5.143	−0.420	0.123
	Married	−6.877	−22.260, 8.507	−0.299	0.359	−15.722	−38.642, 7.199	−0.342	0.174
Monthly disposable income		2.504	−2.469, 7.478	0.269	0.303	9.987	2.160, 17.814	0.343	0.014
Education (RG = Education ≤ 9 years)	9–12 years	−4.797	−15.981, 6.387	−0.209	0.378	−3.888	−20.576, 12.800	−0.085	0.641
	>12 years	−8.771	−22.259, 4.716	−0.335	0.188	−2.318	−25.588, 20.953	−0.027	0.842
Number of insurances policies		−12.708	−27.714, 2.299	−0.507	0.092	−14.289	−25.427, −3.151	−0.331	0.013

Statistically significant mean difference (*p* < 0.05). B: Regression coefficient; CI: confidence intervals; *β*: standardized coefficients; Adj-R^2^: adjusted R squared.

## Data Availability

Not applicable.

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
