# Peer review of "Monthly Disposable Income Is a Crucial Factor Affecting the Quality of Life in Patients with Knee Osteoarthritis"

_healthcare, 2021, doi:10.3390/healthcare9121703_

Round 1

Reviewer 1 Report

This manuscript reports a cross-sectional study in a small group of elderly subjects with knee osteoarthritis in comparison to controls without knee osteoarthritis focussing on dimensions of the SF-36 and their relation to biographic data. Although the topic fits very well to the journal submitted, several shortcomings in the introduction, study design, description of methodology, result reporting and discussion leads to the recommendation of reject to resubmit this paper.

Although the English is generally acceptable, revision by a natural speaker is recommended for minimising ambiguous wordings such as ambulatory VAS-score. Literally, this means that the VAS-score is walking around, while a visual analogue scale (VAS) is employed to capture the pain experienced in walking. The respective item in the WOMAC-score is labelled “pain during walking”.

The introduction section suffers from the condition, that most of the statements are not supported by the provided references. Epidemiological data must be related to the primary epidemiological studies, not to secondary papers that themselves quote the original findings insufficiently. This applies to the references 2, 3, 4 and 7.  Reference 5 dealing with the incidence of osteoarthritis is completely misinterpreted as the statement that “in the United States, 70%–90% of older people experience pain and discomfort due to knee OA“ not supported by this article published in 1995. Reference 6 is likewise misinterpreted since data from obese patients with knee osteoarthritis should not be generalised for all osteoarthritis patients. While the perceived health of an individual is an important component of health, neither reference 9 nor reference 10 discuss the availability of assessment tools for the above purpose. Reference 15 deals with the discriminative power of SF-36 items that may vary in subgroups of osteoarthritis patients, not with biographic features of the investigated population.

More adequate references are needed to describe the epidemiology (Osteoarthritis and Cartilage 2018, 26, 1636-1642, British Medical Bulletin 2013; 105, 185–199), disease burden of osteoarthritis (MJA 2004; 180, S11–S17; Lancet 2018; 392: 1789–85 Arthritis Care & Research 2020, 72(2) 193–200), quality- of-life research (Health Qual Life Outcomes (2021) 19,130) and socioeconomic impact [Nat. Rev. Rheumatol. 2014; 10, 437–441; Journal of Health Services Research & Policy 2013, 18(1) 21–27).

Participants

The inclusion criteria are vaguely described. Radiographic osteoarthritis grade 2 or 3 Kellgren-Lawrence is clearly stated. Also, a symptomatic state was required since recruitable patients participated in a rehabilitative intervention. The kind of symptoms (pain, stiffness, restricted range of motion), the minimum intensity of symptoms or the rehabilitation setting (indoor, outdoor) is not reported. There are no criteria for selecting healthy controls. It seems, that health of controls was defined by the absence of symptomatic knee osteoarthritis, since only 7/30 controls did not have a history of a chronic disease.

The variables “low-income household”, “monthly disposable income” and “number of insurance policies” remain undefined. More details of the Public Health System of Taiwan should be reported to understand the financial impact of insurance policies.

The transformation of the original SF-36-scores is not clearly described. The version and the language of SF-36 and its validation must be reported. Also, whether the questionnaire was completed by the patients with or without assistance.  

Statistical analysis

What test was used to compare biographic characteristics? Did the authors check normal distribution prior to the t-test?

Discussion
Since low income is related to poor health status in a manyfold of diseases, cross-referencing other diseases to explain an association between disposable finances and poor health condition in osteoarthritis patients should be avoided, since  these relationship may rather be general and not disease specific.

Author Response

  1. This manuscript reports a cross-sectional study in a small group of elderly subjects with knee osteoarthritis in comparison to controls without knee osteoarthritis focussing on dimensions of the SF-36 and their relation to biographic data. Although the topic fits very well to the journal submitted, several shortcomings in the introduction, study design, description of methodology, result reporting and discussion leads to the recommendation of reject to resubmit this paper.

  • Although the English is generally acceptable, revision by a natural speaker is recommended for minimising ambiguous wordings such as ambulatory VAS- score. Literally, this means that the VAS-score is walking around, while a visual analogue scale (VAS) is employed to capture the pain experienced in walking. The respective item in the WOMAC-score is labelled “pain during walking”.

Response:

Thanks for the reviewer’s comments, the description of the VAS score is as follows.

  • The introduction section suffers from the condition, that most of the statements are not supported by the provided references. Epidemiological data must be related to the primary epidemiological studies, not to secondary papers that themselves quote the original findings insufficiently. This applies to the references 2, 3, 4 and 7.

Reference 5 dealing with the incidence of osteoarthritis is completely misinterpreted as the statement that “in the United States, 70%–90% of older people experience pain and discomfort due to knee OA“ not supported by this article published in 1995. Reference 6 is likewise misinterpreted since data from obese patients with knee osteoarthritis should not be generalised for all osteoarthritis patients.

Response:

Thanks for the reviewer’s comments, regarding the reference, I have made some additions and changes, please review it.

  • While the perceived health of an individual is an important component of health, neither reference 9 nor reference 10 discuss the availability of assessment tools for the above purpose. Reference 15 deals with the discriminative power of SF-36 items that may vary in subgroups of osteoarthritis patients, not with biographic features of the investigated population.

Response:

Thanks for the reviewer’s precious comments, regarding the reference, I have made some additions and changes, please review it.

  • More adequate references are needed to describe the epidemiology (Osteoarthritis and Cartilage 2018, 26, 1636-1642, British Medical Bulletin 2013; 105, 185–199), disease burden of osteoarthritis (MJA 2004; 180, S11–S17; Lancet 2018; 392: 1789–85 Arthritis Care & Research 2020, 72(2) 193–200), quality- of-life research (Health Qual Life Outcomes (2021) 19,130) and socioeconomic impact [Nat. Rev. Rheumatol. 2014; 10, 437–441; Journal of Health Services Research & Policy 2013, 18(1) 21–27).

Response:

Thanks for the reviewer’s comment and suggestion, I have made some supplements to the article and cited the literature you suggested, please review the manuscript.

  1. Participants

  • The inclusion criteria are vaguely described. Radiographic osteoarthritis grade 2 or 3 Kellgren-Lawrence is clearly stated. Also, a symptomatic state was required since recruitable patients participated in a rehabilitative intervention. The kind of symptoms (pain, stiffness, restricted range of motion), the minimum intensity of symptoms or the rehabilitation setting (indoor, outdoor) is not reported. There are no criteria for selecting healthy controls. It seems, that health of controls was defined by the absence of symptomatic knee osteoarthritis, since only 7/30 controls did not have a history of a chronic disease.

Response:

Thanks for the reviewer’s precious comments, the following is a description of the inclusion criteria for subjects and control groups.

  • The variables “low-income household”, “monthly disposable income” and “number of insurance policies” remain undefined. More details of the Public Health System of Taiwan should be reported to understand the financial impact of insurance policies.

Response:

Thanks for the reviewer’s precious comments, the definitions of “low-income household”, “monthly disposable income” and “number of insurance policies” are as follows.

  • The transformation of the original SF-36-scores is not clearly described. The version and the language of SF-36 and its validation must be reported. Also, whether the questionnaire was completed by the patients with or without assistance.

Response:

Thanks for the reviewer’s comments, the version and the language of SF-36 and its validation I cite the literatures.

  1. Statistical analysis

  • What test was used to compare biographic characteristics? Did the authors check normal distribution prior to the t-test?

Response: Thanks for the reviewer’s precious comments. We explain the comparison method of biographic characteristics in the article, and after the Kolmogorov-Smirnov test is performed on the four continuous variables of age, height, weight, and monthly disposable income, these four variables are determined. This variable does not meet the normal distribution, so we changed the result to Mann–Whitney U test for verification, and corrected the p-value value in Table1:

  1. Discussion

  • Since low income is related to poor health status in a manyfold of diseases, cross-referencing other diseases to explain an association between disposable finances and poor health condition in osteoarthritis patients should be avoided, since these relationship may rather be general and not disease specific.

Response:

Thanks for the reviewer’s precious comments, I have made some changes and additions to the content of the article.

Reviewer 2 Report

The authors examined the relationship between disposable income and quality of life with patients with OA compared to healthy control patients. The study was well designed and the results were compelling. My only critique while reading was I was concerned about the difference between causation and correlation in this study. The authors do address this concern at the end of the discussion, stating they cannot determine causation. However, the title suggests causation. I suggest a title edit that more fits the findings of the study so as not to mislead readers.

Author Response

  1. The authors examined the relationship between disposable income and quality of life with patients with OA compared to healthy control patients. The study was well designed and the results were compelling. My only critique while reading was I was concerned about the difference between causation and correlation in this study. The authors do address this concern at the end of the discussion, stating they cannot determine causation. However, the title suggests causation. I suggest a title edit that more fits the findings of the study so as not to mislead readers.

     Response:

      1). First of all, thank you very much for the suggestions made by the reviewers. The    

           title of this research points out that ”Monthly Disposable Income is a Crucial Factor    

           Affecting the Quality of Life in Patients with Knee Osteoarthritis„. It is indeed the   

           conclusion of our research.

      2). The original meaning has been presented by ”Study Limition„.

Reviewer 3 Report

This might be an interesting study showing outcomes on the quality of life (QOL) of elderly people affected by knee osteoarthritis (OA).
Authors found that monthly disposable income considerably affects the QOL of patients with knee OA.
How do authors justify their study and what was the hypothesis of the study?
Was any power analysis performed before the study? How was performed?
What are the strengths of the study? Is the sample size the only limitation of this study?
What are the immediate clinical implications of these findings for orthopedic surgeons?

Author Response

  1. How do authors justify their study and what was the hypothesis of the study?

      Response:

Thanks for the reviewer’s precious comments.

     1). In this cross-sectional study, 30 healthy controls and 60 patients with mild-to-moderate   

          bilateral knee OA aged between 55 and 75 years were enrolled. All participants

          completed a questionnaire containing questions on 10 demographic characteristics and

          the Medical Outcome Study 36-Item Short-Form Health Survey (SF-36), and their

          QOL scores in the eight dimensions of the SF-36 were evaluated. In the OA group,

          significant cor relations were observed between monthly disposable income and

          physical and mental health components. Monthly disposable income was found to

          considerably affect the QOL of patients with knee OA. Monthly disposable income is a

          crucial factor affecting the QOL of patients with bilateral knee OA.

     2). This study assumed that the younger the age, the male, the lower the height, the lighter  

          the weight, the fewer chronic diseases, non-low-income households, the married, the

          higher the monthly disposable income, the more years of education, and the greater the

          number of insurances, the performance of SF36 the better.

  1. Was any power analysis performed before the study? How was performed?

      Response:

      Thanks for the reviewer’s precious comments.

      In this study, based on the suggestion of Cohen (1988) with large of effect size, we set  

      Cohen's d=0.8, significance level α=0.05. Power ranged from 0.8 to 0.9, and

      calculated the number of subjects in each group to be collected 25 ~33 subjects.

  1. What are the strengths of the study? Is the sample size the only limitation of this study?

      Response:

      Thanks for the reviewer’s precious comments.

      1). If medical behavior or medical treatment is also a kind of consumption, according to Keynes's consumption theory:Consumption is a function of disposable income: C = f(Yd). Take the linear consumption function as an example: C = a + bYd

C: Consumption

            a: Basic or Minima Consumption, when Yd = 0,C = a > 0

          b: Marginal Propensity to Consume, MPC

          0 ≤ MPC = dC/dYd ≤ 1, That is, when the monthly disposable income increased 1     

                                                  dollar, the increment in consumption.

          T: Tax

          Y: Income

          Yd: monthly disposable income = Y – T

          It means that when the monthly disposable income is higher, more can be consumed,

          that is, there are more medical methods available.

     2). In addition to sample size, the age and area are also limitations of this study

  1. What are the immediate clinical implications of these findings for orthopedic surgeons?

      Response:

    Thanks for the reviewer’s precious comments. For patients with lower monthly disposable income, it is not necessary to recommend high-cost treatments. Conversely, patients with higher monthly disposable income can, and only then can they (or have the "ability") accept or choose different medical treatment methods.

Round 2

Reviewer 1 Report

The manuscript has substantially improved and all questions raised in the previous version have been sufficiently answered. It is recommended to accept the paper for publication without further amendments.

Reviewer 3 Report

All revisions have been fulfilled.